# Modes of Transmission of Severe Acute Respiratory Syndrome-Coronavirus-2 (SARS-CoV-2) and Factors Influencing on the Airborne Transmission: A Review

**DOI:** 10.3390/ijerph18020395

**Published:** 2021-01-06

**Authors:** Mahdieh Delikhoon, Marcelo I. Guzman, Ramin Nabizadeh, Abbas Norouzian Baghani

**Affiliations:** 1Department of Occupational Health Engineering, School of Public Health, Isfahan University of Medical Sciences, Isfahan, Iran; mdelikhon@yahoo.com; 2Department of Chemistry, University of Kentucky, Lexington, KY 40506, USA; marcelo.guzman@uky.edu; 3Department of Environmental Health Engineering, School of Public Health, Tehran University of Medical Sciences, Tehran, Iran; rnabizadeh@gmail.com

**Keywords:** coronavirus, COVID-19, infection, SARS-CoV-2, pandemic

## Abstract

The multiple modes of SARS-CoV-2 transmission including airborne, droplet, contact, and fecal–oral transmissions that cause coronavirus disease 2019 (COVID-19) contribute to a public threat to the lives of people worldwide. Herein, different databases are reviewed to evaluate modes of transmission of SARS-CoV-2 and study the effects of negative pressure ventilation, air conditioning system, and related protection approaches of this virus. Droplet transmission was commonly reported to occur in particles with diameter >5 µm that can quickly settle gravitationally on surfaces (1–2 m). Instead, fine and ultrafine particles (airborne transmission) can stay suspended for an extended period of time (≥2 h) and be transported further, e.g., up to 8 m through simple diffusion and convection mechanisms. Droplet and airborne transmission of SARS-CoV-2 can be limited indoors with adequate ventilation of rooms, by routine disinfection of toilets, using negative pressure rooms, using face masks, and maintaining social distancing. Other preventive measures recommended include increasing the number of screening tests of suspected carriers of SARS-CoV-2, reducing the number of persons in a room to minimize sharing indoor air, and monitoring people’s temperature before accessing a building. The work reviews a body of literature supporting the transmission of SARS-CoV-2 through air, causing COVID-19 disease, which requires coordinated worldwide strategies.

## 1. Introduction

The rapid outbreak of coronavirus disease 2019 (COVID-19) caused by the severe acute respiratory syndrome coronavirus 2 (SARS-CoV-2) that is spread via airborne, droplet, contact, and fecal–oral transmission modes constitutes a worldwide and disruptive challenge to societies [1,2]. Both symptomatic and asymptomatic carriers have been recognized to transmit the SARS-CoV-2 virus via various modes of transmission [3,4,5], causing severe respiratory infections among patients receiving medical care in hospitals [6,7,8,9,10,11]. Hence, the COVID-19 outbreak has been declared a global public health emergency of international concern by the World Health Organization (WHO) [12,13,14]. The original source and the way of transmission to humans are the subject of current studies [13,15,16,17]. It was stated in January 2020 that the virus of SARS-CoV-2 causes COVID-19, creating infections typical for a second-class infectious disease, but that management measures for a first-class infectious (dangerous) disease are required [12]. At least a 70% genetic similarity of SARS-CoV-2 to the original Severe Acute Respiratory Syndrome (SARS), as well as it being a subspecies of Sarbecovirus, has been reported [18,19].

Coronaviruses such as SARS, endemic human coronaviruses (HCoV), and Middle East Respiratory Syndrome (MERS) can remain biologically active for ≤9 days on the surface of plastic, metal, and glass [19]. Because the incubation time of SARS-CoV-2 vary from 2 to 14 days, infected persons not displaying symptoms and unaware of their infection are active carriers for transmission to others [12,13,15,16]. Generic symptoms of COVID-19 disease are common to SARS and MERS infections including fatigue, nausea, diarrhea, vomiting, breathing difficulty, dry cough, fever, and bilateral lung infiltration in severe cases [13,18,20].

On 7 February 2020, COVID-19 cases existed in twenty-six countries around the world: Canada, the United States, Australia, India, Sri-Lanka, Cambodia, Thailand, Vietnam, Malaysia, Singapore, Taiwan, South Korea, Japan, Philippines, Nepal, United Arab Emirates, Russia, Italy, Germany, Sweden, Finland, Belgium, Spain, France, and the United Kingdom [9,10]. Most of the confirmed cases were travelers connected to Wuhan or other Chinese cities; however, locally transmitted cases outside of China had also been reported [10]. Moreover, respiratory transmission of the novel coronavirus by airborne mechanisms from aerosols and droplets gained consideration [21,22,23,24,25]. SARS-CoV-2 can be spread via near contact with an infected person coughing or sneezing and emitting small droplets [21,26,27,28]. Transmission via droplets occurs when a SARS-CoV-2 carrier talking, coughing, and sneezing emits droplets with the virus that can expose other people’s nose, mouth, and eyes causing infection [24]. Those generated droplets are relatively large and cannot stay in the air for long time and deposit relatively rapidly [24,29,30,31]. Hence, near contact with an infected person is the main way for virus intake causing infection [9,20,22,23,24,30,31,32].

In addition, indirect long-distance transmission (aerosol transfer) is possible and can happen [9,33,34,35]. The virus emitted in small particles may be transferred a long-distance [9]. In other words, airborne transmission by small droplet nuclei or particulates including the virus is favored due to their extended time suspended in air, and their farther distance traveled from the emitter [24]. For example, concentrated airborne droplets (fragmented) are released through respiration or from sneezing by patients with SARS-CoV-2 virus, reaching several meters away to also remain in suspension for about 30 min, and once settled onto objects can still be biologically active for a few days [22,36].

Because there is no specific therapeutic drug or vaccine for COVID 19 yet [20,37,38,39,40,41,42], it is important to reduce exposure to SARS-CoV-2 [9]. Thus, specific control measures for SARS-CoV-2 transmission are needed [43,44,45,46], such as those listed by the US Centers for Disease Control and Prevention (CDC). The CDC measures include suitable sanitary habits such as utilizing regular hand washing with soap and water, or an alcohol-based hand rub (ABHR) with at least 60% alcohol, using protective masks in public, restricting social contacts (restricting human to human transmission pathway), cleaning and disinfecting surfaces continuously, staying at home as much as possible, and applying proper ventilation such as displacement ventilation (DV) to decrease infection risk [6,9,13,20,47,48,49]. Wearing protective masks in public is perhaps the most cost-effective way for preventing to SARS-CoV-2 spreading. However, temporary shortages in supply of facemask have occurred during the outbreak of COVID-19 due to limitations in production and import/export of KN95/N95 respirators and medical protective masks [10,23,50].

### Aim

To the best of our knowledge, this is the first study reviewing the recent literature for SRAS-CoV-2 transmission by grouping them into four structured sections that including an introduction, a methodology section, the results and discussion, and the concluding remarks. The work explains common symptoms of COVID-19, modes of transmission of SARS-CoV-2, concentration, and infectious dose for airborne transmission of SARS-CoV-2, the influencing factors on the airborne transmission of SARS-CoV-2, and protection approaches to prevent the transmission of SARS-CoV-2. Finally, several proposed protocols for SARS-CoV-2 prevention and a future perspective of the current situation are presented.

## 2. Method

### 2.1. Design

This work originated from a systematic review to provide a narrative synthesis performed according to the Preferred Reporting Items for Systematic Reviews and Meta-analysis (PRISMA) statement [51].

### 2.2. Search Strategy

To perform this work, papers appearing in international databases (Medline/PubMed, Scopus, Google Scholar, Web of Sciences, Science Direct, and Embase) were identified by the following keywords: coronavirus, coronavirus disease 2019, COVID-19, personal protective equipment, transmission, airborne, contact, fecal–oral, droplet, outbreak, beta-CoVs, respiratory, SARS-CoV-2, bioaerosol, aerosol, airborne particle, air, negative pressure, ventilation, air conditioning, social distancing, N95 respirator, mask, alcohol, facemask, ambient air, and indoor air. Due to the high number of articles and to keep the literature up to date, the search was constrained on articles published between 1 March 2020 and 28 December 2020.

### 2.3. Inclusion Criteria

In addition, preference was given to papers in journals that provide information in the field of airborne and droplet transmission of SARS-CoV-2 virus; the effects of different factors such as environmental conditions, negative pressure ventilation (NPV), air conditioning system, displacement ventilation (DV), noninvasive positive pressure ventilation (NIPPV), and high-flow nasal cannula (HFNC) on the airborne transmission of SARS-CoV-2; and effects of protection approaches for transmission of SARS-CoV-2.

### 2.4. Exclusion Criteria

The papers with the following contents were considered to be out of the scope of this review and therefore removed: (i) the epidemiology, virology, and clinical features (imaging features) of SARS-CoV-2; (ii) the transfer of SARS-CoV-2 via wastewater (sewage) and solid waste; and (iii) the effects of COVID-19 on the mental health and quality of people’s life, culture, education, politics, and economy of countries.

## 3. Results

### Results of the Structured Literature Review

First, the titles and abstracts of articles were examined. Then, the investigation was extended to study the full text of related articles as define above. Finally, the selected articles in the field of transmission modes of bioaerosols such as SARS-CoV-2 were studied. As illustrated in Figure 1, the search strategy produced 297 records. The result set was reduced to 215 records following removal of duplicate and non-English language records. The focus of this research was limited to studies on the airborne, droplet, contact, and fecal–oral transmissions of SARS-CoV-2; the effects of environmental conditions (e.g., relative humidity, temperature, and evaporation), NPV, DV, air conditioning system, NIPPV, and HFNC on the airborne transmission of SARS-CoV-2; and protection approaches for transmissions of SARS-CoV-2. Based on a review of the abstracts for relevance, we excluded 83 records. After reading the methods and results sections of the 132 remaining reports, we excluded six reports that did not meet our inclusion criteria: three were the epidemiology and clinical features of SARS-CoV-2 and three were wastewater (sewage) studies. Thus, 126 individual reports were included in our work. The search findings are shown in Figure 1 as a PRISMA flow chart.

## 4. Discussion

### 4.1. Common Symptoms of Coronavirus Disease 2019 (COVID-19)

Usual human coronaviruses, containing alpha and beta coronaviruses, commonly create mild to moderate upper respiratory tract infections (URTI) similar to the common cold [47,52,53]. Meanwhile, the novel coronavirus (SARS-CoV-2) causes severe respiratory illness (lungs and upper respiratory tract) and even death in some patients (Figure 2) [15,16,17,20,27,47,54]. It should be noted that SARS-CoV-2 belongs to beta-CoVs, and its size is between 60 and 140 nm [55] and approximately 120 nm in diameter [56]. Moreover, some studies reported that SARS-CoV-2 has a zoonotic origin [15,16,17,20,27,47,54].

Common signs of this disease are fever, dyspnea, cough, and fatigue, while less usual signs consist of diarrhea, sputum production, hemoptysis, and headache [10,16,57]. In addition, previous studies demonstrated that COVID-19 can cause pneumonia, kidney failure, severe acute respiratory syndrome, and even death [16,58,59]. Besides, Deng et al. (2020) reported that the main symptoms experienced by forty-five fatal cases included shortness of breath, chest pain, fever, and cough [37]. Various symptoms of COVID-19 that can be expressed from 2 to 14 days after infection including fever, cough, and shortness of breath and can become severe enough to result fatal to some patients [38,60,61,62].

Chhikara et al. reported that patients with COVID-19 displayed all symptoms of this disease within 2–7 days [57], and the WHO reported that some patients have a mild sore throat, fever, cough, and a runny nose, while others suffer severe symptoms with breathing difficulties and pneumonia leading to death [20].

### 4.2. Modes of Transmission of SARS-CoV-2

#### 4.2.1. Droplet Transmission

Respiratory transmission of the SARS-CoV-2 can occur by droplet transmissions [5,22,23,24,32]. Transmission via droplets happens when carriers of the virus talking, coughing, breathing, singing, and sneezing emit bioaerosol particles that can reach other people’s nose, mouth, and eyes causing infection (Figure 2) [24]. In addition, Holland et al. reported that, similar to influenza, mumps, *Haemophilus influenzae*, whooping cough, and rubella, coronaviruses are transmitted by bioaerosol droplets [24]. Furthermore, Abkarian et al. and Chong et al. described how flows generated during speaking, singing, breathing, and laughing by asymptomatic and presymptomatic humans contribute to the spread of SARS-CoV-2 virus [4,5]. In addition, experiments and simulations were performed to quantify how exhaled air is transferred during talking and stressed that phonetic characteristics add complexity to the airflow dynamics, i.e., plosive sounds, such as certain words starting with the letter “P” in English, create intense vortical structures that act as “puffs” and quickly reach 1 m distances [4]. Moreover, speech corresponding to a train of such puffs can make a conical, turbulent, jet-like flow and generate directed transport more than 2 m away in 30 s of conversation [4]. Bourouiba et al. illustrated that, depending on the patient’s physiological composition (droplets of mucosalivary liquid emissions associated with hot and moist air) and meteorological conditions (humidity and temperature), the gas cloud and its payload of pathogen-bearing droplets in any size can be transmitted up to 8 m [64]. Along the way, droplets of any size can settle out or evaporate at rates that depend not only on their size but also on the degree of turbulence and velocity of the gas cloud, along with the properties of the ambient environment (temperature, humidity, and air flow) [64,65,66]. Chong et al. reported that smaller respiratory droplets (<10 µm) have the tendency to be taken via the turbulent puff and travel together with the fluid. This can result in an increase of smaller relative velocities and less evaporation owing to the reduction of convective effects, while larger respiratory droplets falling out from the puff settle more quickly than the surrounding fluid [5].

According to pervious work, bioaerosol droplets with different particle size are generated by people [32,67]. Produced bioaerosol droplets in common breathing are in the 0.8–2 μm range, whereas for speaking bimodal distributions in the 16–125 and 0.8–7 μm ranges [32]. Likewise, coughing creates particles from 0.6 to 16 and 40 to 125 μm, while sneezing mostly from 7 to 125 μm [32]. Additionally, previous works stated that produced bioaerosol droplets in coughing and sneezing can range from <1 to >500 or <2000 μm (or many aerosols) [67,68,69].

Stariolo provided two models for the time that generated bioaerosol droplets remain in undisturbed air [23]. In Model 1, droplets with size higher than 200 µm are determined not to be a health risk if social distancing about 1.5–3 m is practiced to avoid SARS-CoV-2 transmission from expiratory activities such as coughing, talking, and sneezing [23,32,49,70]. For droplets <100 µm, a second model of motion in viscous air was employed [23]. In addition, other authors stated that droplet transmission occurred from particles larger than 5 µm, which can underdo gravitational settling on surfaces [9,71] (Figure 2). Because gravitational settling of these particles occurs within 1 m, a cautionary distance of 2 m is considered an appropriate preventive measure (Figure 3) [9,45,71]. Furthermore, to diminish the numbers of COVID-19 cases, it is necessary to slow down the transmission of the life-threatening SARS-CoV-2 virus via virus-laden droplets from screaming, shouting, speaking, breathing, singing, sneezing, or even coughing. Hence, for reducing infections via such respiratory droplets, governments around the world have offered the so-called “2-m distance rule” or “6-foot rule” [1,5].

Many studies assumed that generated bioaerosol droplets from people (e.g., droplets produced by talking, coughing, breathing, singing, and sneezing) are relatively large (0.6–2000 μm), remain in air for short time, and are deposited relatively rapidly [24,29,30,31]. However, those works recognized the transmission of SARS-CoV-2 by bioaerosol droplets can proceed.

#### 4.2.2. Contact Transmission

The contact transmission of *SARS-CoV-2* can happen from surfaces and objects contaminated with SARS-CoV-2 (Figure 2d), where viruses can remain viable to cause infection for several hours and even days [71,72]. For instance, according to Figure 4, SARS-CoV-2 can survive 2–9 days on the surfaces and objects such as plastic (≥4 days), metal (5 days), wood (4–5 days), gowns (2 days), latex gloves (≤8 h), paper (1–5 days), and glass (4 days) [73].

For example, Fiorillo et al. reported that various coronaviruses survive on infected surfaces for up to nine days, and disinfecting infected surfaces with 62–71% ethanol for 1 min or 0.1% sodium hypochlorite can eliminate coronaviruses [74].

In other words, efforts to prevent SARS-CoV-2 transmission by contact with the infected surface or objects (fomite transmission) are also important [9]. For example, the SARS-CoV-2 could also be attached to clothing and distributed by contact [9,26,27,28], putting at high risk the front-line healthcare workers (HCWs) and communities at large [35,71].

#### 4.2.3. Fecal–Oral Transmission

Special attention should be paid to the possibility of fecal transmission, as *SARS-CoV-2* has recently been observed in the feces of infected people [9]. Furthermore, Yang and Wang (2020) reported that the SARS-CoV-2 has been observed in the feces of patients, which may pollute aquatic environments and also cause a fecal–oral route of transmission [13,75]. In addition, previous work reported that the bioaerosols released from the infectious feces from COVID-19 patients can create infection [9,13,75]. According to Xu et al. and Liao et al., airborne aerosol transmission of SARS-CoV-2 is possible from sewage including fecal matter from carrier persons [9,76]. Airborne aerosol transmission of SARS-CoV-2 is described in Section 4.2.4.

#### 4.2.4. Airborne Transmission

Although bioaerosol was a common term applied for microorganisms suspended in indoor or ambient air [77,78,79,80], in medical terminology, bioaerosols are airborne particles with biological matter [32,81]. The mechanism of airborne transmission involves small droplet nuclei (fluid of pathogenic droplets) or particulates including the virus that can stay suspended in air for an extended period of time (≥2 h) and can move farther distances from carriers of SARS-CoV-2 [24]. For example, concentrated airborne droplets (fragmented) that are released during respiration or sneezing by SARS-CoV-2 carriers, may move several meters (up to 8 m), stay suspended for about 30 min, and remain viable on objects for a few days [22,36]. In addition, Lednicky et al. reported that viable SARS-CoV-2 was isolated from air samples gathered 2–4.8 m away from patients [82].

According to the WHO, airborne transmission is possible by virions in droplet nuclei <5 μm that can be present in air for extended periods of time traveling distances >1 m [29]. Airborne transmission by particles <5 µm can distribute SARS-CoV-2 through the air for several hours [71,83,84]. Furthermore, Lee (2020) reported that the minimum size of a respiratory particle that can include SARS-CoV-2 is computed to be about 4.7 µm [85].

The WHO recently suggested that airborne transmission of SARS-CoV-2 can occur in special conditions and settings such as “cardiopulmonary resuscitation (CPR), endotracheal intubation, bronchoscopy, open suctioning, positive pressure ventilation, high-flow nasal cannula (HFNC), and manual ventilation before intubation”, which in this situations may create aerosols [29,71,86]. The question then arises: “Is it likely that the SARS-CoV-2 spreads by air?” [45].

The WHO has not established the importance of airborne transmission of SARS-CoV-2. Thus, official measures by governments such as that of Italy still recommend keeping a distance of just 1 m (red zone) [87,88] because long-range spreading of SARS-CoV-2 is ignored. Hence, how can the daily increase in the number of COVID-19 cases in the United States and Italy be justified? [45]. The complexity to establish the presence of virions in air may have delayed establishing this transmission route [45]. Importantly, it is a well-known that SARS-CoV-1 was spread via aerosol particles in air [32]. Furthermore, many similarities between SARS-CoV-1 and SARS-CoV-2 and their symptoms were reported, suggesting the high likelihood for SARS-CoV-2 airborne transmission [45,89]. The COVID-19 pandemic could then share a common airborne transmission mechanism with tuberculosis and varicella, which is more effective than droplet transmission [24,32,90].

Hence, the final answer to the question is that, due to the increasing trends of infection with SARS-CoV-2 around the world, and knowing the basic science of viral infection spread, there are strong reasons for researchers to believe that SARS-CoV-2 is spreading via the air [45]. In other words, airborne transmission is an important mechanism for the fast spreading of SARS-CoV-2 [24,32,45,91]. In this context, the existing compelling evidence calls for reconsidering only direct contact transmission as the main way of a viral infection. The airborne transmission route should not be ignored by government officials but become a serious concern during the epidemic [45,92].

### 4.3. Concentration and Infectious Dose for Airborne Transmission of SARS-CoV-2

The load of SARS-CoV-2 RNA in suspended particles for potential airborne transmission was investigated by Liu et al. for patient and medical staff areas of Renmin hospital of Wuhan University and Wuchang Fangcang Field hospital and public areas in Wuhan. They collected 35 samples and discriminated by size (total suspended particle, size segregated, and deposition aerosol) [46]. The aerosol fraction was divided into five sizes that included: 0–0.25 μm, 0.25–0.5 μm, 0.5–1 μm, 1–2.5 μm, and ≥2.5 μm [46]. The results show that the RNA concentration of SARS-CoV-2 aerosol in various areas of Fangcang hospital were 1–9 copies/m^3^ [46]. The highest concentration of SARS-CoV-2 RNA for both hospitals was about 19 copies/m^3^ and found in the patient mobile toilet room [46]. Furthermore, considerable levels of SARS-CoV-2 RNA was detected in two definite size ranges: submicron particles (0.25–1.0 µm) with concentrations 9 and 40 copies/m^3^ and supermicron particles (>2.5 µm) with concentrations 7 and 9 copies/m^3^ [46]. Deposition aerosol samples gathered from intensive care unit (ICU) rooms of Renmin hospital showed SARS-CoV-2 RNA concentrations from 31 to 113 copies/m^2^ per hour, and the maximum deposition rates were found in the hindrance-free corner of the room (3 m from the patient’s bed) [46].

Guzman pointed out that the submicron and supermicron aerosols carrying pathogens can coexist owing to the different production routes and resuspension creating supermicron particles that included pathogens from the objects such as protective equipment and shoes producing secondary aerosols [32,46]. Supermicron particles including SARS-CoV-2 can settle on the nasopharyngeal area, pass via the mucous membranes, and replicate to continue distributing to other tissues and organs, including the lungs [32,93]. Furthermore, 2.5–5 μm particles can settle on the trachea, whereas particles ≤2.5 μm (fine particles) and ≤0.1 μm (ultrafine particles) can reach the lungs tissues and settle in the alveolar ducts and sacs [32,94]. Such particles can contribute to the direct transmission of SARS-CoV-2; otherwise, it would be difficult to explain how ten individuals from three families eating at the same restaurant in Guangzhou (China) became sick with COVID-19, unless there was a contribution to the spreading of exhaled particles <5 µm in size by air-conditioned ventilation [32,94]. In addition, in agreement with the perspective of Guzman, Cervino et al. reported that healthcare workers may transmit the viable virus from the floor of one ward to another on their shoes [1,32,95].

Particular care must be taken of reemitted copies of the virus that had settled on the protective clothes of medical workers and floor surface during long time, which can be aerosolized by moving objects and stepping on floors [32,46]. Thus, following recommendations for sanitization of surfaces, protective equipment and clothing is crucial for reducing aerosol transmission of SARS-CoV-2 [32,46]. The infectious dose of SARS-CoV-2 is believed to be low (1 × 10^2^ to 1 × 10^3^ particles) due to the fast distribution ability of the virus around the world [32,45,96,97]. Hence, it is imperative for researchers to evaluate how low the concentration of virus, for different particle sizes, may lead to infection [32,91]. For instance, scanning electron microscopy (SEM) works of gathered aerosol samples demonstrated the virion integrity of SARS-CoV-2 may be kept more than 16 h suspended in the air [32,91].

### 4.4. Influencing Factors on the Airborne Transmission of SARS-CoV-2

#### 4.4.1. Environmental Conditions

Various parameters such as relative humidity, temperature, and evaporation affect the transmission of SARS-CoV-2 [32,45,48,71,98,99,100]. Yao et al. described a decrease of SARS-CoV-2 transmission at high temperature and low relative humidity for ambient ozone levels [48]. While large droplets including virus are deposited on near surface, if liquid content of the same virus-laden particles is loss to air by evaporation, those resulting smaller airborne particles can move via air current rather than by gravitational settling [45]. The evaporation of water droplets <10 µm occurs in less than 1 s, leaving nuclei that may include virions suspended in air for several hours (e.g., for transmission in hospital rooms and interior of airplanes) [23]. In addition to simple diffusion in quiet air, the convection process is surely another main method for airborne infection. It has been described that, even in the presence of convective flows, the virus can distribute to a large room before settling or scattering on surfaces [23]. Furthermore, Chong et al. reported how variable environmental conditions such as a raise in relative humidity from 50% to 90% can extend the lifetime of droplets with 10 µm diameter and affect airborne transmission of SARS-CoV-2 over large distances [5]. They also stressed that smaller droplets live longer and travel farther than large droplets [5]. By tracking Lagrangian statistics, Ng et al. reported that cold and humid environments reduce the ability of air to keep water vapor, leading to the respiratory vapor puff to supersaturate [101]. As a result, the supersaturated vapor field drives the growth of droplets that are caught and transported within the humid puff [101]. Hence, droplets smaller than 10 µm can reach farther distances when the weather is cold and humid [101]. In addition, Chaudhuri et al. described that, as the droplet evaporation time specifies the infection rate constant, environmental conditions affect the COVID-19 outbreak growth rates [102]. Although warm weather can reduce the growth rates, rigid social enforcement measures are also needed together with contact tracing, quarantining, and the widespread use of face masks to control the COVID-19 pandemic [102].

Cervino et al. reported the remarkable persistence of the virus at low temperature [95]. For example, after fourteen days at 4 °C, the viral titer was reduced only by 0.7 logarithmic units [95]. Instead, by increasing the incubation temperature to 70 °C, the virus was no longer detectable after only 5 min [95]. Moreover, changes in pH did not affect the stability of SARS-CoV-2, as studied in the range from pH 3 to 10 [95].

#### 4.4.2. Negative Pressure Ventilation (NPV), Displacement Ventilation (DV), Air Conditioning System, Noninvasive Positive Pressure Ventilation (NIPPV), and High-Flow Nasal Cannula (HFNC)

Santarpia and coauthors collected air samples from eleven isolation rooms at the University of Nebraska Medical Center (including thirteen COVID-19 patients) for examining viral shedding from patients. They showed that SARS-CoV-2 was poured into the air as expired particles, during toilet flushing and via contact with fomites. They concluded that SARS-CoV-2 can be transmitted directly from patients by droplets and indirectly from air and polluted contacts [103]. The same study detected SARS-CoV-2 RNA using real-time reverse transcriptase–polymerase chain reaction (RT-PCR) and reported the mean concentration of 2.86 copies/L of air (63.2% positive) in patient rooms. The previous observation indicated the transmission of aerosols could have occurred even in the absence of cough or aerosol producing ways because samples were collected in rooms with negative pressure ventilation (NPV) [103]. They interpreted that air turbulence can lead to suspend the droplets with large size for a long time in isolation rooms [103]. However, further Tan et al., Liu et al., Vukkadala et al., Cook, Holland et al., and Fathizadeh et al. discussed an opposite scenario and mentioned that, to prevent SARS-CoV-2 transmission, patients should be kept in negative pressure rooms as much as possible [24,46,71,73,104,105]. For example, Tan et al., the CDC, and Holland et al. declared that, to keep particles with SARS-CoV-2 from escaping the areas with infected patients, rooms under negative pressure should be used [24,104,106].

Air conditioning systems have been considered to be just as important as negative pressure ventilation (NPV) systems. Air conditioning of contaminated rooms must be exercised under negative pressure mode due to the fact that those rooms apply lower air pressure to permit outside air into the divided environment, keeping harmful particles from escaping to ambient air [45,46,71,73].

Previous works described that patients rooms with NPV and high air-exchange rates in different parts of Renmin Hospital were useful for reducing airborne transmission of SARS-CoV-2 [46,73]. In addition, Chiu and co-authors mentioned that endoscopies in cases of suspected or confirmed SARS-CoV-2 should be done in NPV room [107]. However, Cook et al. stated that the high air-exchange rates in the rooms were more important than whether the rooms were under negative or positive pressure and they recommended that particles should be prevented from settling in places with a low rate of air exchange or no ventilation [71]. Furthermore, Bhagat et al. performed a study about different forms of ventilation, including mixing ventilation, natural and mechanical displacement ventilation, and wind-driven ventilation [108]. Importantly, if displacement ventilation is designed properly to promote vertical stratification, it can reduce indoor warm air pollutants close to the ceiling [108]. Hence, displacement ventilation should be the most effective for diminishing the exposure risk of SARS-CoV-2 and can be simply and cheaply installed via vents or fans at the top of the space [108]. Instead, mixing ventilation disperses indoor air throughout the space and cannot support any potentially clean zones [108].

Miller et al. evaluated the aerosol generation with noninvasive positive pressure ventilation (NIPPV) and the high-flow nasal cannula (HFNC) in contrast to 6 L per minute using low-flow nasal cannula in healthy volunteers [109]. The results show that NIPPV and HFNC did not raise SARS-CoV-2 aerosol emissions in contrast to low-flow nasal cannula in healthy volunteers [109]. Li and co-authors (2020) reported that production and distribution of SARS-CoV-2 by HFNC can occur [86]. In addition, Li et al. explained that, to control the production and distribution of SARS-CoV-2 aerosol from COVID-19 patients, surgical masks can be put on the patient’s face and prong it with a high-flow nasal cannula (HFNC) [86]. The protection approaches for transmissions of SARS-CoV-2 is described in Section 4.5.

### 4.5. Protection Approaches for Transmissions of SARS-CoV-2

Due to the high prevalence of novel coronavirus in the world, precise control measures for SARS-CoV-2 are needed [43,44,45,46]. Liao et al. reported that, during a choir practice (singing), forty-five persons of a sixty-member choir group were exposed to SARS-CoV-2, and two members died [67]. According to past work, the mean number of people an individual transfers SARS-CoV-2 to is 2 ≤ R_0_ ≤ 2.5 [32]. Thus, local governments suggest controlling the COVID-19 pandemic by practicing social distancing and staying at home (virus source control) [23,32,38,61,63,94,110,111,112,113,114,115].

Efforts to control ventilation related spreading of the virus in restaurants, cruise ships, nursing homes, schools, kindergartens, offices, shops, and public transport are essential to controlling the pandemic [45,116]. Particular individuals with high levels of infection should be restricted from human-to-human contact and polluting the environment to protect others [13,60]. A simple and cost-effective way for preventing SARS-CoV-2 distribution is by everyone to wear protective masks in public places [10,23,45,50,117,118]. While surgical masks can decrease the spread of COVID-19 [12], N95 masks are needed by workers in healthcare facilities combined with gowns and goggles [12,50]. Additionally, Fiorillo et al. suggested that health care workers must be protected from the contact with the patient’s aerosol with filtering facepiece particles masks (FFP) of type FFP2 and FFP3 [74] of similar performance to N95 and N99 masks in the United States, respectively. In the case of surgical mask, their filtration efficiency only protects the user against droplet nuclei of size larger than 2 μm [65]. In general, face coverings are more effective for decreasing the direct ejection of breath and bioaerosols away from the user [108]. Disinfection of surfaces, i.e., with suitable detergents, along with frequent hand washing and using of appropriate personal protective equipment are important preventive measures to decrease SARS-CoV-2 transmission [74].

In addition to proper ventilation, cleaning methods such as mechanical air filters, ultra-violet germicidal irradiation (UVGI), high-efficiency particulate air (HEPA) filters and ion generators, and even other chemical methods such as alcohol, hydrogen peroxide, and sodium hypochlorite cab be used for deactivating coronaviruses [9,78,119]. For example, isopropyl and ethyl alcohol have been applied as disinfectants in healthcare settings in recent years [120]. Previous studies demonstrated that 60–70% solutions of ethyl alcohol can deactivate murine norovirus, Ebola virus, and many coronaviruses [47,120,121]. However, due to the airborne distribution of SARS-CoV-2, disinfection of ambient air in communities and cities appears more challenging as high levels of alcohol and other disinfectants in air are a potentially danger for people [12]. Hence, the use of chemical disinfection methods and face masks appear as the most appropriate measures for limiting the spreading of this pandemic [12,45,59].

Although some countries are developing and starting to implement the use of vaccines at present, many nations can only implement preventive measures and effective health responses by government officials, medical doctors, and the public to restrict the spread of COVID-19. In addition to other modes of transmission of SARS-CoV-2, person-to-person transmission of SARS-CoV-2 is a serious threat to public health due to its fast dissemination in public gatherings [122]. Some important steps needed to prevent the spreading of COVID-19 are:(1)Isolation of the affected persons, including asymptomatic carriers and travelers from affected countries [122](2)Applying travel restrictions from and to infected countries [122](3)Avoiding social gatherings and events [122](4)Extending knowledge of the public awareness about COVID-19 [122](5)Wearing of personal protective equipment, especially masks, such as N95, FFP2, and FFP3, along with hand washing and using of appropriate protective clothes by healthcare workers [12,50,74,122](6)Ensuring measures to boost the immune system such as consuming nutrient with suitable vitamins such as C and E [122,123](7)Keeping social distancing (about 1.5–3 m) [23,32,49,70,122](8)Applying proper ventilation such as displacement ventilation [103,108]

Summarizing, to limit airborne and droplet transmissions of SARS-CoV-2, the following are recommended: (i) adequate ventilation or good air-conditioning systems for rooms [45,46,78,94,103]; (ii) appropriate disinfection of toilets [46]; (iii) the use of open space with minimum crowds [46]; (iv) utilization of negative pressure rooms [24,103]; (v) screening and testing of individuals suspected of carrying SARS-CoV-2 [46,124,125]; (vi) monitoring temperature at close space such as restaurant [94]; (vii) keeping a cautionary distance to others [32,45,71,98]; (viii) avoiding the direct airflow of others [45,126]; (ix) reducing the number of individuals in a common place for minimizing sharing indoor air [45,127]; and (x) minimizing the number of shared devices and objects with other people [45,71,72].

## 5. Conclusions

COVID-19 has become a global health concern creating severe respiratory tract infections and other complications. The pathways of probable transmission of SARS-CoV-2 considered in this work include direct contact, airborne, fecal–oral, and droplet transmission routes. The influence of negative pressure ventilation and air conditioning systems on the airborne transmission was discussed. The major findings of this work are:The airborne transmission of SARS-CoV-2 is an important contributor to fast spreading of the associated disease.Droplet transmission occurs from particles >5 µm, which can settle on surfaces under gravitational settling and do not move more than 1 m. Particles <5 µm can stay suspended for an extended period of time (≥2 h) and travel longer distances (up to 8 m) through simple diffusion and convection mechanisms.The environmental ambient conditions can affect airborne transmission of SARS-CoV-2 over larger distances.The droplets <10 µm in size can be transferred larger distant when the weather is cold and humid.The persistence of the virus is remarkable at a low temperature (4 °C), and, by raising the temperature to 70 °C, the virus was no longer detectable after 5 min.Although warm weather can slow down the growth rates of SARS-CoV-2, rigid social enforcement and other measures such as the widespread use of face masks are still needed to control the COVID-19 pandemic.Coordinated measures among the public and private sectors are needed to control this disease at the national and international level. Joint solutions from different experts, not only in the biomedical sciences, but also in the environment, chemistry, physics, public health, and areas covering transportation, immigration, economic affairs, and education, are required.SARS-CoV-2 can also be resuspended from floor surfaces and from protective clothes and shoes of medical workers in indoor environments. Indeed, direct contact with fomites is not the only way for causing viral infection with SARS-CoV-2. As explained above, SARS-CoV-2 is distributed via air. Thus, disinfection of surfaces and protective equipment before removal is needed together with frequent hand washing.The various coronaviruses survive on surfaces for up to nine days, and they can be eliminated by disinfection with 62–71% ethanol for 1 min or 0.1% sodium hypochlorite.Healthcare workers must be provided with N95, FFP2, or FFP3 masks combined with gowns and goggles.Overall, control measures such as using high adequate ventilation, rooms with negative pressure ventilation, practicing social distancing, and wearing N95 and even surgical facemasks are potentially suggested to reduce the SARS-CoV-2 airborne transmission.

### 5.1. Future Perspective

Future experimental work is certainly needed to expand the current understanding of the findings reported above and on other modes of SARS-CoV-2 transmission by water, wastewater (sewage), and infectious solid waste. In addition, in-depth studies should be performed to explain the effect of air pollution, e.g., PM_2.5_, PM_10_, SO_2_, NO_2_, Pb, VOCs, and CO, on the airborne and fomite transmission of SARS-CoV-2 to diverse populations. Studying the influence of sunny, rainy, and smog days on the transmission modes of SARS-CoV-2 should be of interest to characterize different scenarios for airborne and fomite transmissions.

### 5.2. Capsule

This work is novel in that there are no reports to our knowledge of the modes of transmission of severe acute respiratory syndrome-coronavirus-2 (SARS-CoV-2) and factors influencing on the airborne transmission. The results of this work have broad implications for people around the world owing to the pervasiveness of Coronavirus Disease 2019 (COVID-19).

## Figures and Tables

**Figure 1 ijerph-18-00395-f001:**
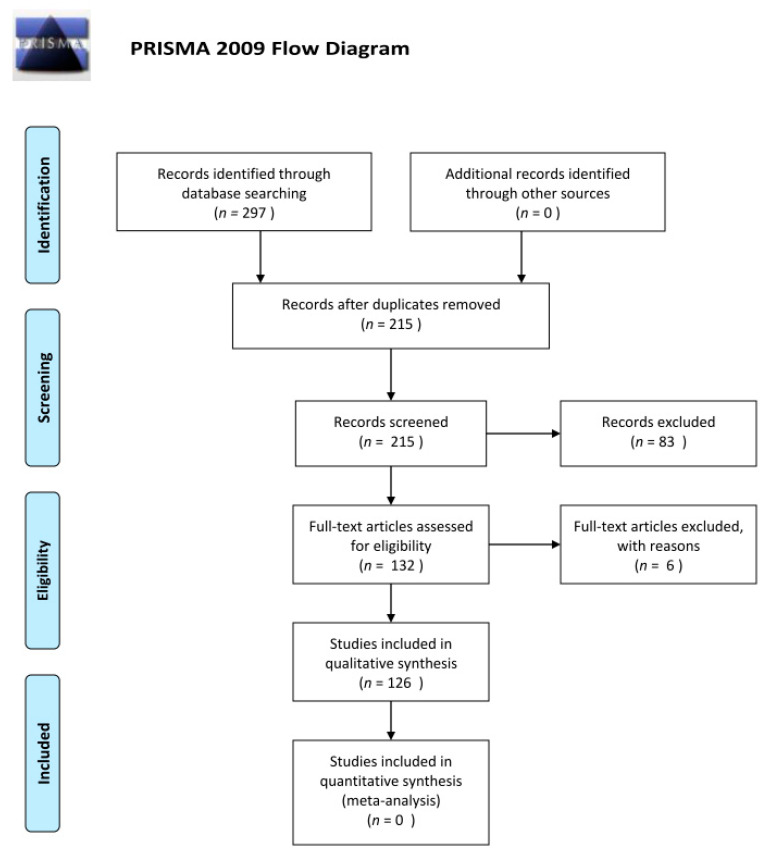
PRISMA 2009 flow diagram of the structured literature review.

**Figure 2 ijerph-18-00395-f002:**
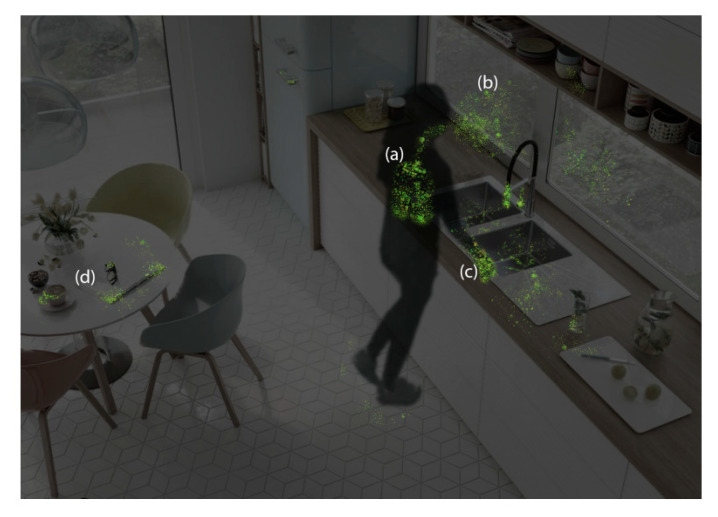
Conceptualization of SARS-CoV-2 deposition: (**a**) once infected with novel coronavirus, viral particles get in the lungs and upper respiratory tract; (**b**) transmission of SARS-CoV-2 via droplets and aerosolized viral particles when COVID-19 patients talking, coughing, breathing, singing, and sneezing and can distribute on the other people’s nose, mouth, eyes, clothes, and nearby surroundings; (**c**) SARS-CoV-2 from the mouth and nose is often observed on the hands; and (**d**) SARS-CoV-2 could be distributed to often touched surfaces and objects [63].

**Figure 3 ijerph-18-00395-f003:**
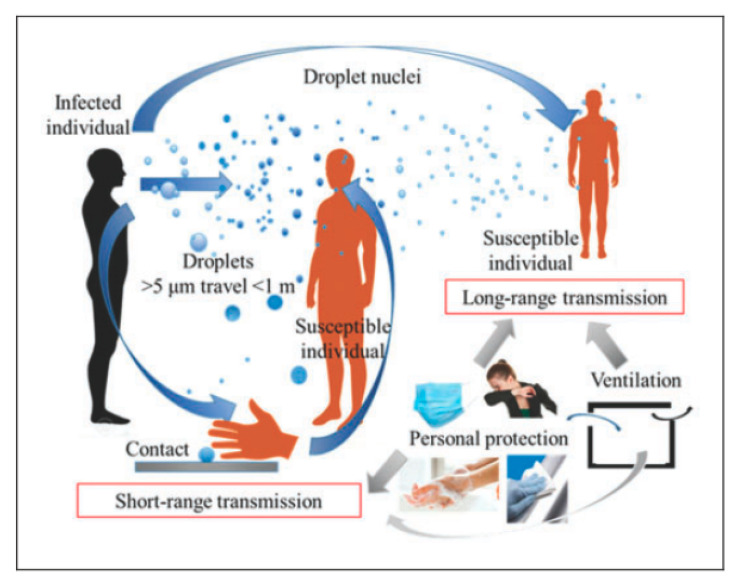
The droplet transmission of SARS-CoV-2 with particles >5 µm [9].

**Figure 4 ijerph-18-00395-f004:**
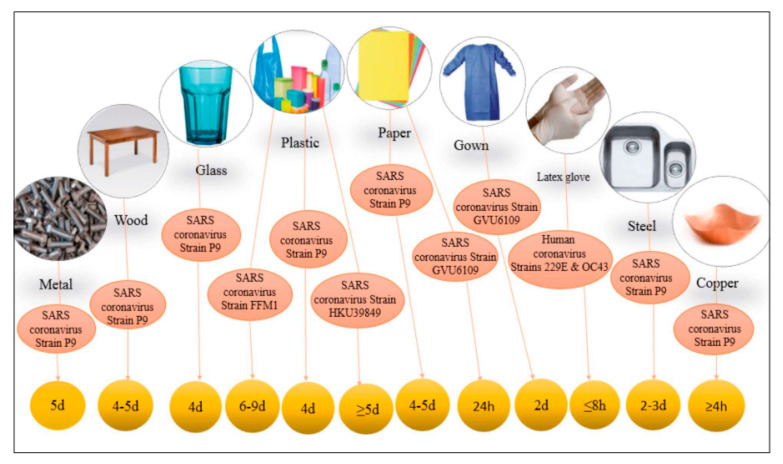
Survival of SARS-CoV-2 on different surfaces [73].

## Data Availability

Not applicable as no new data were created in this review.

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
