# Peer review of "Modes of Transmission of Severe Acute Respiratory Syndrome-Coronavirus-2 (SARS-CoV-2) and Factors Influencing on the Airborne Transmission: A Review"

_ijerph, 2021, doi:10.3390/ijerph18020395_

Round 1
Reviewer 1 Report
See the attachment.

Author Response
Dear Professor Andjelka Jovanovic
Editor in Chief, International Journal of Environmental Research and Public Health
Thank you very much for your response to our manuscript entitled: “Modes of transmission of severe acute respiratory syndrome -coronavirus-2 (SARS-CoV-2) and factors influencing on the airborne transmission: a review” (Manuscript ID: ijerph-1053741). The comments have been carefully accounted for and the responses to the comments are shown subsequently in blue font. The corresponding corrections have been added to the manuscript file and highlighted in yellow color. We thank the referees for thoughtful reviews and the comments that helped us improve the manuscript.
Best wishes,
Sincerely
Abbas Norouzian Baghani
Corresponding author
Question 1) The first few sentences about the origin of the outbreak are too controversial. That is not something we should comment too much as still a lot of debates over this topic. I suggest the authors to tune the sentences further down.
Answer: Thank you for your very careful review of our paper, and for the comments. We eliminated the first few sentences about the origin of the outbreak in the manuscript and added another sentences. Please see lines 12-14 in Abstract and lines: 30-37 in Introduction sections.
Hence: We remove these sentences in Abstract and Introduction sections.
Original:
Abstract section
Despite of the special importance of novel coronavirus (SARS-CoV-2) spreading mechanisms, airborne, droplet, contact, and fecal-oral transmissions are still under discussion.
Introduction section
The outbreak of SARS-CoV-2 determined in Wuhan (China) in December 2019 and then distributed to Guizhou province on January of 2020 that caused coronavirus disease 2019 (COVID-19) among people [1-6]. During the outbreak of SARS-CoV-2, severe respiratory infections were observed among patients [1-6]. Some of those patients had a history of presenting or working in the wholesale fish and seafood market [7].
Hence: We added these sentences in Abstract (Lines: 12-14) and Introduction sections (Lines: 30-37).
Revised:
Abstract section
The multiple modes of SARS-CoV-2 transmission including airborne, droplet, contact, and fecal-oral transmissions that caused coronavirus disease 2019 (COVID-19) contribute a public threat to the lives of people worldwide.
Introduction section
The rapid outbreak of coronavirus disease 2019 (COVID-19) caused by the severe acute respiratory syndrome coronavirus 2 (SARS-CoV-2) that is spread via airborne, droplet, contact, and fecal-oral transmission modes constitutes a worldwide and disruptive challenge to societies [8, 9]. Both symptomatic and asymptomatic carriers have been recognized to transmit the SARS-CoV-2 virus via various modes of transmission [10-12] causing severe respiratory infections among patients receiving medical care in hospitals [1-6]. Hence, the COVID-19 outbreak has been declared a global public health emergency of international concern by the World Health Organization (WHO) [13-15]. The original source and way of transmission of SARS-CoV-2 to humans is the subject of current studies [14, 16-18].
Question 2) Line 61-63, when the authors talk about the transmission route of respiratory droplets, I think the classical work by Wells, Am. J. Epidemiol., 20 (1934) should also be cited. I understand that the authors would like to mainly cite papers in 2020. However, given that the authors have also cite some very relevant papers published before 2020, I suggest that this very relevant paper should also be cited.
Answer: Thank you for your very careful review of our paper, and for the comments. We are happy to use and cite these very relevant papers published before 2020 such as:
Wells, W. F., On air-borne infection: study ii. droplets and droplet nuclei. American Journal of Epidemiology 1934, 20, (3), 611-618.
Wells, W. F.; Stone, W. R., On air-borne infection: study iii. viability of droplet nuclei infection. American Journal of Epidemiology 1934, 20, (3), 619-627.
Please see lines 58-60 and lines 201-203.
Hence:
Those generated droplets are relatively large and cannot stay in the air for long time and deposit relatively rapidly [19-22]. Hence, near contact with an infected person is the main way for virus intake causing infection [4, 19, 21-26].
And:
Please see lines 58-60 and lines 201-203.
Many studies assumed that generated bioaerosol droplets from people (e.g., droplets produced by talking, coughing, breathing, singing, and sneezing) are relatively large (from 0.6 to 2000 μm), remain in air for short time, and are deposited relatively rapidly [19-22].
Question 3) Line 78-79, about the ventilation, there is recent review by Bhagat et al., JFM, 903 (2020) which is also very relevant. Different ventilation strategies have been mentioned in their work, eg mixed ventilation, displacement ventilations. The authors should comment on that to provide a more comprehensive review.
Answer: Thank you for your very careful review of our paper, and for the comments. We now use and cite this relevant paper. We added relevant sentences in the section “4.4.2. Negative Pressure Ventilation (NPV), Displacement Ventilation (DV), Air Conditioning System, Noninvasive Positive Pressure Ventilation (NIPPV), and High-Flow Nasal Cannula (HFNC) ”.
Please see lines 360-366.
Hence:
Furthermore, Bhagat et al. (2020) performed a study about different forms of ventilation, including mixing ventilation, natural and mechanical displacement ventilation, and wind-driven ventilation [27]. Importantly, if displacement ventilation is designed properly to promote vertical stratification, it can reduce indoor warm air pollutants close to the ceiling [27]. Hence, displacement ventilation should be the most effective for diminishing the exposure risk of COVID-19, and can be simply and cheaply installed via vents or fans at the top of the space [27]. Instead, mixing ventilation disperses indoor air throughout the space and cannot support any potentially clean zones [27].
We added the following additional sentences in the section of “4.5. Protection Approaches for Transmissions of SARS-CoV-2”:
Please see lines 392-393.
In general, face coverings are more effective for decreasing the direct ejection of breath and bioaerosols away from the user [27].
Furthermore, we added some final notes related to those studies in the Conclusion section.
Such as point 8) in Conclusion section.
Please see line 442.
8) Applying proper ventilation such as displacement ventilation [27, 28].
The referred studies were also used in section “1. Introduction”.
Please see lines 73-76.
Hence:
using protective masks in public, restricting social contacts (restricting human to human transmission pathway), cleaning and disinfecting surfaces continuously, staying at home as far as possible, and applying proper ventilation such as displacement ventilation (DV) to decrease infection risk [1, 4, 14, 23, 29-31].
Question 4) Line 148-149: About the droplet transmission route, there are recently significant progress on understanding the such route from fluid mechanics point of view. For example, 1) the work by Abkarian et al., PNAS, 117 (2020) about the jet-like transport in speech. 2) the work by Chong et al. (2020) arXiv:2008.01841 about the extended lifetime of respiratory droplets. 3) Bourouiba, JAMA, 323 (2020) about the respiratory droplets in turbulent gas clouds. 4) Balachandar et al. IJMF, 132 (2020) gives the overview of airborne transimission. I think these are also the very relevant new findings in this field.
Answer: Thank you for your very careful review of our paper, and for the comments. We now use these suggested relevant papers in section of “4.2.1. Droplet Transmission”.
Hence, we added these sentences in this section. Please see lines 157-179.
4.2.1. Droplet Transmission
Respiratory transmission of the SARS-CoV-2 can occur by droplet transmissions [12, 19, 24-26]. Transmission via droplets happens when carriers of the virus talking, coughing, breathing, singing, and sneezing emit bioaerosol particles that can reach to other people's nose, mouth, and eyes causing infection (Figure 2) [19]. In addition, Holland et al. reported that like influenza, mumps, haemophilus influenzae, whooping cough, and rubella, coronaviruses are transmitted by bioaerosol droplets [19]. Furthermore, Abkarian et al. and Chong et al. described how flows generated during speaking, singing, breathing, and laughing by asymptomatic and presymptomatic humans contribute to the spread of SARS-CoV-2 virus [11, 12]. In addition, experiments and simulations were performed to quantify how exhaled air is transferred during talking and stressed that phonetic characteristics add complexity to the airflow dynamics, i.e. plosive sounds such certain words starting with the letter “P” in English, create intense vortical structures that act like “puffs” and quickly reach 1 meter distances [11]. Moreover, speech corresponding to a train of such puffs can make a conical, turbulent, jet-like flow and generates directed transport more than two meters away in 30 second of conversation [11]. Related work by Bourouiba et al. illustrated that depending on the patient's physiological composition (droplets of mucosalivary liquid emissions associated with hot and moist air) and meteorological conditions (humidity and temperature), the gas cloud and its payload of pathogen-bearing droplets in any size can be transmitted up to 8 meters far [32]. Along the way, droplets of any size can settle out or evaporate at rates that depend not only on their size, but also on the degree of turbulence and velocity of the gas cloud, along with the properties of the ambient environment (temperature, humidity, and air flow) [32-34]. Chong et al. reported that smaller respiratory droplets (< 10 µm) have the tendency to be taken via the turbulent puff and travel together with the fluid. This can result in an increase of smaller relative velocities and less evaporation owing to the reduction of convective effects, while larger respiratory droplets falling out from the puff settle faster than the surrounding fluid [12].
And: Please see lines 193-198.
Furthermore, to diminish the numbers of COVID-19 cases, it is necessary to slow down the transmission of the life-threatening SARS-CoV-2 virus via virus-laden droplets from screaming, shouting, speaking, breathing, singing, sneezing, or even coughing. Hence, for reducing infections via such respiratory droplets, governments around the world have offered the so-called “2-meter distance rule” or “6-foot rule” [8, 12].
In addition, we used and cited the recommended paper in section of “4.2.4. Airborne Transmission”:
Please see lines 261-262.
In other words, airborne transmission is an important mechanism for the fast spreading of SARS-CoV-2 [19, 26, 35, 36].
In addition, we used and cited relevant paper published in section of “4.5. Protection Approaches for Transmissions of SARS-CoV-2”:
Please see lines 388-395.
Additionally, Fiorillo et al. suggested that health care workers must be protected from the contact with the patient’s aerosol with filtering facepiece particles masks of type FFP2 and FFP3 [37] of similar performance to N95 and N99 masks in the United States, respectively. In the case of surgical mask, their filtration efficiency only protects the user against droplet nuclei of size larger than 2 μm [33]. In general, face coverings are more effective for decreasing the direct ejection of breath and bioaerosols away from the user [27]. Disinfection of surfaces, i.e. with suitable detergents, along with frequent hand washing and using of appropriate personal protective equipment are important preventive measures to decrease SARS-CoV-2 transmission [37].
In addition, we used and cited the relevant paper in section of “4.4.1. Environmental Conditions”:
Please see lines 314-318.
Furthermore, Chong et al. reported how variable environmental conditions such as a raise in relative humidity from 50% to 90% can extend the lifetime of droplets with 10 µm diameter and affect airborne transmission of SARS-CoV-2 over large distances [12]. The same work also stressed that smaller droplets live longer and travel farther than large droplets [12].
Question 5) Line 269-270: About the influence of environmental conditions on transmission, there are also recent progress 1) Ng et al. (2020) arXiv:2011.01515 about the fate of respiratory droplets in different ambient condition. 2) Chaudhuri et al. (2020) arXiv:2004.10929 about how to model the effect of ambient condition on droplets.
Answer: Thank you for your very careful review of our paper, and for the comments. We have now used and cited both relevant papers in section of “4.4.1. Environmental Conditions”:
Please see lines 318-326.
By tracking Lagrangian statistics, Ng et al. reported that cold and humid environments reduce the ability of air to keep water vapor, leading to the respiratory vapor puff to supersaturate. As a result, the supersaturated vapor field drives the growth of droplets that are caught and transported within the humid puff. Hence, droplets smaller than 10 µm can reach farther distances when the weather is cold and humid [38]. In addition, Chaudhuri et al. described that as the droplet evaporation time specifies the infection rate constant, environmental conditions affect the COVID-19 outbreak growth rates. Although warm weather can reduce the growth rates, rigid social enforcement measures are also needed together with contact tracing, quarantining and the widespread use of face masks to control the COVID-19 pandemic [39].
References:
- Chen, S.; Yang, J.; Yang, W.; Wang, C.; Bärnighausen, T., COVID-19 control in China during mass population movements at New Year. The Lancet 2020.
- Ping, K., Epidemiologic Characteristics of COVID-19 in Guizhou, China. medRxiv 2020.
- WHO, W. H. O. Health workers exposure risk assessment and management in the context of COVID-19 virus: interim guidance, 4 March 2020; World Health Organization: 2020.
- Xu, C.; Luo, X.; Yu, C.; Cao, S.-J., The 2019-nCoV epidemic control strategies and future challenges of building healthy smart cities. In SAGE Publications Sage UK: London, England: 2020.
- Wu, H.; Huang, J.; Zhang, C. J.; He, Z.; Ming, W.-k., Facemask shortage and the coronavirus disease (COVID-19) outbreak: Reflection on public health measures. medRxiv 2020.
- Rodriguez-Morales, A. J.; Sánchez-Duque, J. A.; Botero, S. H.; Pérez-Díaz, C. E.; Villamil-Gómez, W. E.; Méndez, C. A.; Verbanaz, S.; Cimerman, S.; Rodriguez-Enciso, H. D.; Escalera-Antezana, J. P., Preparación y control de la enfermedad por coronavirus 2019 (COVID-19) en América Latina. ACTA MEDICA PERUANA 2020, 37, (1), 3-7.
- Jahanbin, K.; Rahmanian, V., Using twitter and web news mining to predict COVID-19 outbreak. 2020.
- Guzman, M. I., An overview of the effect of bioaerosol size in coronavirus disease 2019 transmission. The International journal of health planning and management 2020.
- Daniela, D. A.; Gola, M.; Letizia, A.; Marco, D.; Fara, G. M.; Rebecchi, A.; Gaetano, S.; Capolongo, S., COVID-19 and Living Spaces challenge. Well-being and Public Health recommendations for a healthy, safe, and sustainable housing. 2020.
- Li, H.; Wang, Y.; Ji, M.; Pei, F.; Zhao, Q.; Zhou, Y.; Hong, Y.; Han, S.; Wang, J.; Wang, Q., Transmission routes analysis of SARS-CoV-2: a systematic review and case report. Frontiers in cell and developmental biology 2020, 8, 618.
- Abkarian, M.; Mendez, S.; Xue, N.; Yang, F.; Stone, H. A., Speech can produce jet-like transport relevant to asymptomatic spreading of virus. Proceedings of the National Academy of Sciences 2020, 117, (41), 25237-25245.
- Chong, K. L.; Ng, C. S.; Hori, N.; Yang, R.; Verzicco, R.; Lohse, D., Extended lifetime of respiratory droplets in a turbulent vapour puff and its implications on airborne disease transmission. arXiv preprint arXiv:2008.01841 2020.
- Xiao, Y.; Torok, M. E., Taking the right measures to control COVID-19. The Lancet Infectious Diseases 2020.
- Yang, C.; Wang, J., A mathematical model for the novel coronavirus epidemic in Wuhan, China. Mathematical Biosciences and Engineering, 2020, 17, (3), 2708-2724.
- WHO, W. H. O. Considerations for quarantine of individuals in the context of containment for coronavirus disease ( COVID-19): interim guidance, 29 February 2020; World Health Organization: 2020.
- CDC, C. f. D. C. a. P., Centers for Disease Control and Prevention. 2019 Novel Coronavirus (2019-nCoV) Situation Summary. Available from: https://www.cdc.gov/coronavirus/2019-nCoV/summary.html. Accessed 3 February 2020. 2020.
- RCP, R. C. P. Novel Coronavirus 2019 | Rubbermaid Commercial Products- ©2020 Rubbermaid Commercial Products LLC 8900 Northpointe Executive Park Drive, Huntersville, NC 28078.
- Bonilla-Aldana, D. K.; Cardona-Trujillo, M. C.; García-Barco, A.; Holguin-Rivera, Y.; Cortes-Bonilla, I.; Bedoya-Arias, H. A.; Patiño-Cadavid, L. J.; Paniz-Mondolfi, A.; Zambrano, L. I.; Dhama, K., MERS-CoV and SARS-CoV Infections in Animals: A Systematic Review and Meta-Analysis of Prevalence Studies. 2020.
- Holland, M.; Zaloga, D. J.; Friderici, C. S., COVID-19 Personal Protective Equipment (PPE) for the emergency physician. Visual journal of emergency medicine 2020, 19, 100740.
- WHO, W. H. O. Modes of transmission of virus causing COVID-19: implications for IPC precaution recommendations: scientific brief, 27 March 2020; World Health Organization: 2020.
- Wells, W. F., On air-borne infection: study ii. droplets and droplet nuclei. American Journal of Epidemiology 1934, 20, (3), 611-618.
- Wells, W. F.; Stone, W. R., On air-borne infection: study iii. viability of droplet nuclei infection. American Journal of Epidemiology 1934, 20, (3), 619-627.
- WHO, W. H. O., The COVID-19 risk communication package for healthcare facilities. 2020.
- Rowan, N. J.; Laffey, J. G., Challenges and solutions for addressing critical shortage of supply chain for personal and protective equipment (PPE) arising from Coronavirus disease (COVID19) pandemic–Case study from the Republic of Ireland. Science of The Total Environment 2020, 138532.
- Stariolo, D. A., COVID-19 in air suspensions. arXiv preprint arXiv:2004.05699 2020.
- Guzman, M., Bioaerosol Size Effect in COVID-19 Transmission. https://www. preprints. org/manuscript/202004.0093/v1/download. 2020.
- Bhagat, R. K.; Davies Wykes, M. S.; Dalziel, S. B.; Linden, P. F., Effects of ventilation on the indoor spread of COVID-19. Journal of Fluid Mechanics 2020, 903, F1.
- Santarpia, J. L.; Rivera, D. N.; Herrera, V.; Morwitzer, M. J.; Creager, H.; Santarpia, G. W.; Crown, K. K.; Brett-Major, D.; Schnaubelt, E.; Broadhurst, M. J., Transmission potential of SARS-CoV-2 in viral shedding observed at the University of Nebraska Medical Center. MedRxIV 2020.
- Millán-Oñate, J.; Rodriguez-Morales, A. J.; Camacho-Moreno, G.; Mendoza-Ramírez, H.; Rodríguez-Sabogal, I. A.; Álvarez-Moreno, C., A new emerging zoonotic virus of concern: the 2019 novel Coronavirus (COVID-19). Infectio 2020, 24, (3).
- Yao, M.; Zhang, L.; Ma, J.; Zhou, L., On airborne transmission and control of SARS-Cov-2. Science of The Total Environment 2020, 139178.
- Kumar, P.; Hama, S.; Omidvarborna, H.; Sharma, A.; Sahani, J.; Abhijith, K. V.; Debele, S. E.; Zavala-Reyes, J. C.; Barwise, Y.; Tiwari, A., Temporary reduction in fine particulate matter due to ‘anthropogenic emissions switch-off’ during COVID-19 lockdown in Indian cities. Sustainable Cities and Society 2020, 62, 102382.
- Bourouiba, L., Turbulent gas clouds and respiratory pathogen emissions: potential implications for reducing transmission of COVID-19. Jama 2020, 323, (18), 1837-1838.
- Balachandar, S.; Zaleski, S.; Soldati, A.; Ahmadi, G.; Bourouiba, L., Host-to-host airborne transmission as a multiphase flow problem for science-based social distance guidelines. In Elsevier: 2020.
- Balachandar, S.; Soldati, A., Multiphase flow community must have a role in predicting host-to-host airborne contagion. International Journal of Multiphase Flow 2020.
- Morawska, L.; Cao, J., Airborne transmission of SARS-CoV-2: the world should face the reality. Environment International 2020, 105730.
- Fears, A. C.; Klimstra, W. B.; Duprex, P.; Hartman, A.; Weaver, S. C.; Plante, K.; Mirchandani, D.; Plante, J.; Aguilar, P. V.; Fernandez, D., Comparative dynamic aerosol efficiencies of three emergent coronaviruses and the unusual persistence of SARS-CoV-2 in aerosol suspensions. medRxiv 2020.
- Fiorillo, L.; Cervino, G.; Matarese, M.; D’Amico, C.; Surace, G.; Paduano, V.; Fiorillo, M. T.; Moschella, A.; La Bruna, A.; Romano, G. L., COVID-19 Surface Persistence: A Recent Data Summary and Its Importance for Medical and Dental Settings. International Journal of Environmental Research and Public Health 2020, 17, (9), 3132.
- Ng, C. S.; Chong, K. L.; Yang, R.; Li, M.; Verzicco, R.; Lohse, D., Growth of respiratory droplets in cold and humid air. arXiv preprint arXiv:2011.01515 2020.
- Chaudhuri, S.; Basu, S.; Kabi, P.; Unni, V. R.; Saha, A., Modeling ambient temperature and relative humidity sensitivity of respiratory droplets and their role in Covid-19 outbreaks. arXiv preprint arXiv:2004.10929 2020.

Reviewer 2 Report
Dear Authors,
Your manuscript is really interesting and well conducted.
Due to the covid Pandemic the manuscript is really current and need to be published as soon as possible.
Please in keyword section use only medical subject headings (MeSHs word) for a better indexing.
In introduction section please better state the aim of the article and if possible create a subsection "Aim"
This is a review (not systematic), but try to follow a template as PRISMA statement for a better organization.
In Your research You missed some inherent and important results as:
1. Fiorillo, L.; Cervino, G.; Matarese, M.; D’Amico, C.; Surace, G.; Paduano, V.; Fiorillo, M.T.; Moschella, A.; La Bruna, A.; Romano, G.L., et al. COVID-19 Surface Persistence: A Recent Data Summary and Its Importance for Medical and Dental Settings. In Int J Environ Res Public Health, 2020; Vol. 17, p 3132.
2. Cervino, G.; Fiorillo, L.; Surace, G.; Paduano, V.; Fiorillo, M.T.; De Stefano, R.; Laudicella, R.; Baldari, S.; Gaeta, M.; Cicciù, M. SARS-CoV-2 Persistence: Data Summary up to Q2 2020. In Data, 2020; Vol. 5, p 81.
You need to describe results of these manuscript in Your research.
Please state some proposed protocols for Sars-Cov-2 prevention in discussion section and state at the end (in conclusion) some bullet points with main results and future perspective of the study
Thank You
Author Response
Dear Professor Andjelka Jovanovic
Editor in Chief, International Journal of Environmental Research and Public Health
Thank you very much for your response to our manuscript entitled: “Modes of transmission of severe acute respiratory syndrome -coronavirus-2 (SARS-CoV-2) and factors influencing on the airborne transmission: a review” (Manuscript ID: ijerph-1053741). The comments have been carefully accounted for and the responses to the comments are shown subsequently in blue font. The corresponding corrections have been added to the manuscript file as highlighted with yellow color. We thank the referees for thoughtful reviews and the comments that helped us improve the manuscript.
Best wishes,
Sincerely
Abbas Norouzian Baghani
Corresponding author
Question 1) Dear Authors, Your manuscript is really interesting and well conducted. Due to the covid Pandemic the manuscript is really current and need to be published as soon as possible. Please in keyword section use only medical subject headings (MeSHs word) for a better indexing.
Answer: Thank you for your very careful review of our paper, and for the comments. We edited the keywords as indicated below. Please see line 27.
Original:
Keywords: Coronavirus; COVID-19; airborne transmission; SARS-CoV-2; negative pressure ventilation.
Revised:
Keywords: Coronavirus; COVID-19; infection; SARS-CoV-2; pandemic.
MeSH Heading : Pandemics
Tree Number(s) N06.850.290.200.600
Unique ID D058873
RDF Unique Identifier http://id.nlm.nih.gov/mesh/D058873
Annotation coordinate IM with specific disease / epidemiol (IM) & specify geographic location if pertinent
Scope Note Epidemics of infectious disease that have spread to many countries, often more than one continent, and usually affecting a large number of people.
Previous Indexing Disease Outbreaks (1967-2010)
Public MeSH Note 2011; see DISEASE OUTBREAKS 2006-2010
History Note 2011; use DISEASE OUTBREAKS 2006-2010
Date Established 2011/01/01 Date of Entry 2010/06/25
MeSH Heading COVID-19
Tree Number(s) C01.748.214
C01.748.610.763.500 C01.925.705.500
C01.925.782.600.550.200.163 C08.381.677.807.500
C08.730.214 C08.730.610.763.500
Unique ID D000086382 RDF Unique Identifier http://id.nlm.nih.gov/mesh/D000086382
Entry Term(s)
2019 Novel Coronavirus Disease
2019 Novel Coronavirus Infection
2019-nCoV Disease
2019-nCoV Infection
COVID-19 Pandemic
COVID-19 Pandemics
COVID-19 Virus Disease
MeSH Heading : SARS-CoV-2
Tree Number(s) B04.820.578.500.540.150.113.968
Unique ID D000086402
RDF Unique Identifier http://id.nlm.nih.gov/mesh/D000086402
Annotation infection = COVID-19
Scope Note A species of BETACORONAVIRUS causing atypical respiratory disease (COVID-19) in humans. The organism was first identified in 2019 in Wuhan, China. The natural host is the Chinese intermediate horseshoe bat, RHINOLOPHUS affinis.
Entry Term(s)
2019 Novel Coronavirus
2019-nCoV
COVID-19 Virus
MeSH Heading: Coronavirus
Tree Number(s) B04.820.578.500.540.150
Unique ID D017934
RDF Unique Identifier http://id.nlm.nih.gov/mesh/D017934
Annotation general or unspecified; prefer ALPHACORONAVIRUS, BETACORONAVIRUS, DELTACORONAVIRUS, GAMMACORONAVIRUS or their specifics; infection = CORONAVIRUS INFECTIONS
Scope Note A member of CORONAVIRIDAE which causes respiratory or gastrointestinal disease in a variety of vertebrates.
Entry Term(s)
Coronavirus, Rabbit
Coronaviruses
Rabbit Coronavirus
MeSH Heading: Infections
Tree Number(s) C01
Unique ID D007239
RDF Unique Identifier http://id.nlm.nih.gov/mesh/D007239
Annotation general only; prefer specifics; /prev = INFECTION CONTROL but see note there
Scope Note Invasion of the host organism by microorganisms or their toxins or by parasites that can cause pathological conditions or diseases.
Entry Term(s)
Infection
Infection and Infestation
Infections and Infestations
See Also Anti-Infective Agents
Question 2) In introduction section please better state the aim of the article and if possible create a subsection "Aim"
Answer: Thank you for your very careful review of our paper, and for the comments. We stated the aims of this work according to PRISMA statement as a subsection "1.1. Aim":
Please see lines 80-88.
“To the best of our knowledge, this is the first study reviewing the recent literature for SRAS-CoV-2 transmission by grouping them in four structured sections that tackle an introduction, a methodology section, describes results and provides a discussion before the concluding remarks. The work explains common symptoms of COVID-19, modes of transmission of SARS-CoV-2, concentration and infectious dose for airborne transmission of SARS-CoV-2, the influencing factors on the airborne transmission of SARS-CoV-2, and protection approaches to prevent the transmission of SARS-CoV-2. Finally, several proposed protocols for SARS-CoV-2 prevention and a future perspective of the current situation are presented.”
Question 3) This is a review (not systematic), but try to follow a template as PRISMA statement for a better organization.
Answer: Thank you for this recommendation for the revision to now use the PRISMA statement for better organization. We indicate in the text that the PRISMA flow diagram in the methods and PRISMA checklist for this work are included:
Please see lines 90-129.
“2.1. Design
This work originated from a systematic review to provide a narrative synthesis performed according to the Preferred Reporting Items for Systematic Reviews and Meta-analysis (PRISMA) statement [1].
2.2. Search strategy
In order to perform this work, papers appearing in international databases (Medline/PubMed, Scopus, Google Scholar, Web of Sciences, Science Direct, and Embase) were identified by the following keywords: coronavirus, coronavirus disease 2019, COVID-19, personal protective equipment, transmission, airborne, contact, fecal-oral, droplet, outbreak, beta-CoVs, respiratory, SARS-CoV-2, bioaerosol, aerosol, airborne particle, air, negative pressure, ventilation, air conditioning, social distancing, N95 respirator, mask, alcohol, facemask, ambient air, and indoor air. Due to the high number of articles and to keep the literature up to date, the search was constrained to articles published between March 1 and December 28, 2020.
2.3. Inclusion criteria
In addition, preference was given to papers in journals that provide information in the field of airborne and droplet transmission of SARS-CoV-2 virus, the effects of different factors such as environmental conditions, NPV, air conditioning system, displacement ventilation (DV), NIPPV, and HFNC on the airborne transmission of SARS-CoV-2, and effects of protection approaches for transmission of SARS-CoV-2.
2.4. Exclusion criteria
The papers with the following contents were considered to be out of the scope of this review and therefore removed: i) the epidemiology, virology, and clinical features (imaging features) of SARS-CoV-2, ii) the transfer of SARS-CoV-2 via wastewater (sewage) and solid waste iii) the effects of COVID-19 on the mental health and quality of people’s life, culture, education, politics, and economy of countries.
- Results
3.1. Results of the Structured Literature Review
First the titles and abstracts of articles were examined. Then, the investigation was extended to study the full text of related articles as define above. Finally, the selected articles in the field of transmission modes of bioaerosols such as SARS-CoV-2 were studied. As illustrated in Figure 1, the search strategy produced 297 records. The result set was reduced to 215 records following removal of duplicate and non-English language records. The focus of this research was limited to studies on the airborne, droplet, contact, and fecal-oral transmissions of SARS-CoV-2 and the effects of environmental conditions (e.g., relative humidity, temperature and evaporation), NPV, DV, air conditioning system, NIPPV, and HFNC on the airborne transmission of SARS-CoV-2 and protection approaches on for transmissions of SARS-CoV-2. Based upon review of the abstracts for relevance, we excluded 83 records. After reading the methods and results sections of the 132 remaining reports, we excluded 6 reports that did not meet our inclusion criteria: 3 were the epidemiology and clinical features of SARS-CoV-2 and 3 were wastewater (sewage) studies. Thus, 126 individual reports were included in our work. The search findings are shown in Figure 1, the PRISMA flow chart.
Figure 1. PRISMA 2009 Flow Diagram of the Structured Literature Review.
In addition, we reorganized the structure of the revised manuscript according to the PRISMA statement as far as possible:
- Introduction
1.1. Aim
- Method
2.1. Design
2.2. Search strategy
2.3. Inclusion criteria
2.4. Exclusion criteria
- Results
3.1. Results of the Structured Literature Review
- Discussion
4.1. Common Symptoms of Coronavirus Disease 2019 (COVID-19)
4.2. Modes of Transmission of SARS-CoV-2
4.2.1. Droplet Transmission
4.2.2. Contact Transmission
4.2.3. Fecal-Oral Transmission
4.2.4. Airborne Transmission
4.3. Concentration and Infectious Dose for Airborne Transmission of SARS-CoV-2
4.4. Influencing Factors on the Airborne Transmission of SARS-CoV-2
4.4.1. Environmental Conditions
4.4.2. Negative Pressure Ventilation (NPV), Displacement Ventilation (DV), Air Conditioning System, Noninvasive Positive Pressure Ventilation (NIPPV), and High-Flow Nasal Cannula (HFNC)
4.5. Protection Approaches for Transmissions of SARS-CoV-2
- Conclusions
5.1. Future Perspective
Question 4) In Your research You missed some inherent and important results as:
- Fiorillo, L.; Cervino, G.; Matarese, M.; D’Amico, C.; Surace, G.; Paduano, V.; Fiorillo, M.T.; Moschella, A.; La Bruna, A.; Romano, G.L., et al. COVID-19 Surface Persistence: A Recent Data Summary and Its Importance for Medical and Dental Settings. In Int J Environ Res Public Health, 2020; Vol. 17, p 3132.
- Cervino, G.; Fiorillo, L.; Surace, G.; Paduano, V.; Fiorillo, M.T.; De Stefano, R.; Laudicella, R.; Baldari, S.; Gaeta, M.; Cicciù, M. SARS-CoV-2 Persistence: Data Summary up to Q2 2020. In Data, 2020; Vol. 5, p 81.
You need to describe results of these manuscript in Your research.
Answer: Thank you for your helpful recommendation of these paper we now included for example in section “4.2.2. Contact Transmission”:
Please see lines 211-213.
“Another example, Fiorillo et al. (2020) reported that various coronaviruses survive in infectious surfaces for up to 9 days and by disinfecting infectious surfaces by 62%–71% ethanol for 1 minute or 0.1% sodium hypochlorite can be eliminated coronaviruses [2].”
In section of “4.5. Protection Approaches for Transmissions of SARS-CoV-2”:
Please see lines 388-391.
“Additionally, Fiorillo et al. suggested that health care workers must be protected from the contact with the patient’s aerosol with filtering facepiece particles (FFP) masks of type FFP2 and FFP3 [2] of similar performance to N95 and N99 masks in the United States, respectively.”
In section “4.3. Concentration and Infectious Dose for Airborne Transmission of SARS-CoV-2”:
Please see lines 291-293.
“In addition, in agreement with the perspective of Guzman, Cervino et al. reported that healthcare workers may transmit the viable virus from the floor of one ward to another on their shoes [3-5].”
In section “4.4.1. Environmental Conditions”:
Please see lines 327-331.
“Cervino et al. reported the remarkable persistence of the virus at low temperature [5]. For example, after fourteen days at 4 °C, the viral titer was reduced only by 0.7 logarithmic units [5]. Instead, by increasing the incubation temperature to 70 °C, the virus was no longer detectable after only five minutes [5]. Moreover, changes in pH did not affect the stability of SARS-CoV-2, as studied in the range from pH 3 to 10 [5].”
Question 5) Please state some proposed protocols for Sars-Cov-2 prevention in discussion section and state at the end (in conclusion) some bullet points with main results and future perspective of the study
Answer: Thank you for your recommendation which we incorporated into the revision by proposing some protocols for SARS-CoV-2 prevention in the discussion (section “4.5. Protection Approaches for Transmissions of SARS-CoV-2”) and conclusion sections. Some bullet points of this study and future perspective are also stated in the conclusion section:
Please see lines 406-422.
“Although some countries are developing and starting to implement the use of vaccines at present, many nations can only implement preventive measures and effective health responses by government officials, medical doctors, and the public to restrict the spread of COVID-19. In addition to other modes of transmission of SARS-CoV-2, person-to-person transmission of SARS-CoV-2 is a serious threat to public health due to its fast dissemination in public gatherings [6]. Some important steps needed to prevent the spreading of COVID-19 in are:
(1) Isolation of the affected persons, including asymptomatic carriers, and travelers from affected countries [6].
(2) Applying travel restrictions from and to infected countries [6].
(3) Avoiding social gatherings and events [6].
(4) Extending knowledge of the public awareness about COVID-19 [6].
(5) Wearing of personal protective equipment, especially masks such as N95, FFP2 and FFP3 along with hand washing and using of appropriate protective clothes by healthcare workers [2, 6-8].
(6) Ensuring measures to boost the immune system such as consuming nutrient with suitable vitamins such as C and E [6, 9].
(7) Keeping social distancing (about 1.5-3 meters) [3, 6, 10-12].
(8) Applying proper ventilation such as displacement ventilation [13, 14].”
As indicated above, we proposed and added some protocols for SARS-CoV-2 prevention in the discussion section; and included some bullet points of this study and future perspective in the “5. Conclusions” section:
Please see lines 432-474.
“COVID-19 has become a global health concern creating severe respiratory tract infections and other complications. The pathways of probable transmission of SARS-CoV-2 considered in this work include direct contact, airborne, fecal-oral, and droplet transmission routes. The influence of negative pressure ventilation and air conditioning systems on the airborne transmission was discussed. The major findings of this work are:
- The airborne transmission of SARS-CoV-2 is an important contributor to fast spreading of the associated disease.
- Droplet transmission occurs from particles > 5 µm, that can settle on surfaces under gravitational settling and do not move more than 1 m far. Particles < 5 µm can stay suspended for an extended period of time (≥ 2 h) for traveling longer distances (up to 8 m) through simple diffusion and convection mechanisms.
- The environmental ambient conditions can affect airborne transmission of SARS-CoV-2 over larger distances.
- The droplets < 10 µm in size can be transferred larger distant when the weather is cold and humid.
- The persistence of the virus was remarkable at a low temperature (4 °C) and by raising the temperature to 70 °C, the virus was no longer detectable after five minutes.
- Although warm weather can slow down the growth rates of SARS-CoV-2, rigid social enforcement and other measures such as the widespread use of face masks is still needed to control the COVID-19 pandemic.
- Coordinated measures among the public and private sectors are needed to control this disease at the national and international level. Joint solutions from different experts, not only in the biomedical sciences, but also in the environment, chemistry, physics, public health, and areas covering transportation, immigration, economic affairs, and education, are required.
- SARS-CoV-2 can also be resuspended from floor surfaces and from protective clothes and shoes of medical workers in indoor environments. Indeed, direct contact with fomites is not the only way for causing viral infection with SARS-CoV-2. As explained above, SARS-CoV-2 is distributed via air. Thus, disinfection of surfaces and protective equipment before removal is needed together with frequent hand washing.
- The various coronaviruses survive on surfaces for up to 9 days and by disinfection with 62%–71% ethanol for 1 minute or 0.1% sodium hypochlorite they can be eliminated.
- The healthcare workers must be provided with N95, FFP2 or FFP3 masks combined with gowns and goggles.
- Overall, control measures such as using high adequate ventilation, rooms with negative pressure ventilation, practicing social distancing, and wearing N95 and even surgical facemasks are potentially suggested to reduce the SARS-CoV-2 airborne transmission.
5.1. Future Perspective
Future experimental work is certainly needed to expand the current understanding of the findings reported above, and on other modes of SARS-CoV-2 transmission by water, wastewater (sewage) and infectious solid waste. In addition, in-depth studies should be performed to explain the effect of air pollution, e.g., PM2.5, PM10, SO2, NO2, Pb, VOCs, and CO, on the airborne and fomite transmission of SARS-CoV-2 to diverse populations. Studying the influence of sunny, rainy and smog days on the transmission modes of SARS-CoV-2 should be of interest to characterize different scenarios for airborne and fomite transmissions.”
References:
- Moher, D.; Liberati, A.; Tetzlaff, J.; Altman, D. G.; The, P. G., Preferred Reporting Items for Systematic Reviews and Meta-Analyses: The PRISMA Statement. PLOS Medicine 2009, 6, (7), e1000097.
- Fiorillo, L.; Cervino, G.; Matarese, M.; D’Amico, C.; Surace, G.; Paduano, V.; Fiorillo, M. T.; Moschella, A.; La Bruna, A.; Romano, G. L., COVID-19 Surface Persistence: A Recent Data Summary and Its Importance for Medical and Dental Settings. International Journal of Environmental Research and Public Health 2020, 17, (9), 3132.
- Guzman, M., Bioaerosol Size Effect in COVID-19 Transmission. https://www. preprints. org/manuscript/202004.0093/v1/download. 2020.
- Guzman, M. I., An overview of the effect of bioaerosol size in coronavirus disease 2019 transmission. The International journal of health planning and management 2020.
- Cervino, G.; Fiorillo, L.; Surace, G.; Paduano, V.; Fiorillo, M. T.; De Stefano, R.; Laudicella, R.; Baldari, S.; Gaeta, M.; Cicciù, M., SARS-CoV-2 persistence: data summary up to Q2 2020. Data 2020, 5, (3), 81.
- Srivastava, N.; Saxena, S. K., Prevention and Control Strategies for SARS-CoV-2 Infection. In Coronavirus Disease 2019 (COVID-19): Epidemiology, Pathogenesis, Diagnosis, and Therapeutics, Saxena, S. K., Ed. Springer Singapore: Singapore, 2020; pp 127-140.
- Xiao, Y.; Torok, M. E., Taking the right measures to control COVID-19. The Lancet Infectious Diseases 2020.
- Adams, J. G.; Walls, R. M., Supporting the Health Care Workforce During the COVID-19 Global Epidemic. JAMA 2020.
- Aman, F.; Masood, S., How Nutrition can help to fight against COVID-19 Pandemic. Pak J Med Sci 2020, 36, (COVID19-S4), S121-S123.
- Stariolo, D. A., COVID-19 in air suspensions. arXiv preprint arXiv:2004.05699 2020.
- Sun, C.; Zhai, Z., The efficacy of social distance and ventilation effectiveness in preventing COVID-19 transmission. Sustainable Cities and Society 2020, 62, 102390.
- Kumar, P.; Hama, S.; Omidvarborna, H.; Sharma, A.; Sahani, J.; Abhijith, K. V.; Debele, S. E.; Zavala-Reyes, J. C.; Barwise, Y.; Tiwari, A., Temporary reduction in fine particulate matter due to ‘anthropogenic emissions switch-off’ during COVID-19 lockdown in Indian cities. Sustainable Cities and Society 2020, 62, 102382.
- Bhagat, R. K.; Davies Wykes, M. S.; Dalziel, S. B.; Linden, P. F., Effects of ventilation on the indoor spread of COVID-19. Journal of Fluid Mechanics 2020, 903, F1.
- Santarpia, J. L.; Rivera, D. N.; Herrera, V.; Morwitzer, M. J.; Creager, H.; Santarpia, G. W.; Crown, K. K.; Brett-Major, D.; Schnaubelt, E.; Broadhurst, M. J., Transmission potential of SARS-CoV-2 in viral shedding observed at the University of Nebraska Medical Center. MedRxIV 2020.

Round 2
Reviewer 2 Report
Authors addressed my comments and now the manuscript has been improved